# Vision-Based Driver’s Cognitive Load Classification Considering Eye Movement Using Machine Learning and Deep Learning

**DOI:** 10.3390/s21238019

**Published:** 2021-11-30

**Authors:** Hamidur Rahman, Mobyen Uddin Ahmed, Shaibal Barua, Peter Funk, Shahina Begum

**Affiliations:** School of Innovation, Design and Engineering, Mälardalen University, 722 20 Västerås, Sweden; mobyen.ahmed@mdh.se (M.U.A.); shaibal.barua@mdh.se (S.B.); peter.funk@mdh.se (P.F.); shahina.begum@mdh.se (S.B.)

**Keywords:** cognitive load, eye-movement, machine learning, non-contact

## Abstract

Due to the advancement of science and technology, modern cars are highly technical, more activity occurs inside the car and driving is faster; however, statistics show that the number of road fatalities have increased in recent years because of drivers’ unsafe behaviors. Therefore, to make the traffic environment safe it is important to keep the driver alert and awake both in human and autonomous driving cars. A driver’s cognitive load is considered a good indication of alertness, but determining cognitive load is challenging and the acceptance of wire sensor solutions are not preferred in real-world driving scenarios. The recent development of a non-contact approach through image processing and decreasing hardware prices enables new solutions and there are several interesting features related to the driver’s eyes that are currently explored in research. This paper presents a vision-based method to extract useful parameters from a driver’s eye movement signals and manual feature extraction based on domain knowledge, as well as automatic feature extraction using deep learning architectures. Five machine learning models and three deep learning architectures are developed to classify a driver’s cognitive load. The results show that the highest classification accuracy achieved is 92% by the support vector machine model with linear kernel function and 91% by the convolutional neural networks model. This non-contact technology can be a potential contributor in advanced driver assistive systems.

## 1. Introduction

Today’s vehicle system is more advanced, faster and safer than before and is on the process to be fully autonomous. Literature shows that most traffic accidents happen by human error [1]. Therefore, theoretically, a well-programmed computer system or autonomous system can reduce the accident rate [2]. Recently, many automobile industries have launched cars with autonomous level 3 and 4; however, in the development process of autonomous vehicles, human drivers must be present in case of failures of autonomous systems or, if necessary, humanitarian assistance [3]. Hence, the necessity of driver monitoring is rapidly increasing in the transportation research community as well as in vehicle industries.

According to National Highway Traffic Safety Administration (NHTSA), about 94% of all observed accidents occurred in 2018 due to the presence of human error [4] such as higher stress [5], tiredness [6], drowsiness [7,8] or higher cognitive load [9]. A report published in 2015 shows that almost 38% of the total road accidents happen due to the driver’s mental distraction [10], which increases cognitive load of the driver. Another driver status called fatigue is the gradually increasing subjective feeling of tiredness of a subject under load. Fatigue can have physical or mental causes and can be manifested in a number of different ways [11].

Generally, three types of parameters are used to monitor drivers’ cognitive load: physiological parameters, vehicle-based parameters and behavioral parameters [12]. Traditionally, physiological parameters are obtained using sensors attached to the driver’s body. However, non-contact-based approach extracts physiological parameters from facial image sequences that capture the color variation of facial skin due to blood circulation caused by cardiac pulses in the cardiovascular system [13,14,15,16]. Recently, non-contact-based heart rate (HR) and heart rate variability (HRV) extraction techniques have been vividly reviewed in [17,18] respectively. Vehicular parameters are also used to classify the driver’s cognitive load, such as in [19,20]. Behavioral parameters—the behavior of the driver, including eye movement, yawning, eye closure, eye blinking, head pose, etc.—are monitored through a digital camera, and the cognitive load of the driver is detected. Robust eye detection and tracking are considered to play a crucial role for driver monitoring based on behavioral measures. The eye-tracking system can be an alternative solution to detect and extract eye movement parameters. Existing eye-tracking systems are either sensor-based or vision-based [21].

For cognitive load monitoring, different parameters have been investigated through different physiological sensors. From the literature (presented in Section 1.1) it is seen that there are not many vision-based contributions in driver monitoring applications. Though a few attempts (i.e., vision-based methods) have been initiated for driver cognitive load monitoring, these are limited to head movement or behavioral activities. In this paper, a vision-based method is implemented to extract eye movement parameters through a driver’s facial images, which is a new and noble contribution in this domain according to our knowledge. Several machine learning (ML) and deep learning (DL) algorithms are deployed to classify the driver’s cognitive load. Here, a single digital camera is used to record the driver’s facial images, and eye pupil positions from each image frame are detected and extracted. Two eye movement parameters, saccade and fixation, are calculated using the eye pupil positions and 13 features are extracted manually. However, in this study, subject fatigue is not considered for cognitive load classification [11]. Five ML algorithms, support vector machine (SVM) [22], logistic regression (LR) [23], linear discriminant analysis (LDA) [24], k-nearest neighbor (k-NN) [25] and decision tree (DT) [26], are deployed for cognitive load classification. Further, three DL architectures: convolutional neural networks (CNN) [27], long-short-term-memory (LSTM) [28,29] and autoencoder (AE) [30] are designed both for automatic feature extraction from raw eye movement signals and for classification. Additionally, combined DL + ML approaches, i.e., CNN + SVM and AE + SVM, are used for feature extraction and classification. In addition, a commercial eye tracker is simultaneously used as a reference sensor. Finally, the performance of the cognitive load classification is evaluated through several statistical measurements: the accuracy, *F*_1_-score, sensitivity and specificity of the camera system are compared with the reference system. To observe the significant difference between the ML and DL algorithms, two statistical significance tests, Wilcoxon’s test and Delong’s test, are conducted. Comparisons between the systems are observed in terms of raw extracted signals, extracted features and different windowing.

In rest of the paper, Section 1.1 presents state-of-the-art, Section 2 describes materials and methods that include the data collection procedure, feature extraction, case formulation and experimental procedure; Section 3 presents experimental works and results, and a discussion is included in Section 4. Finally, Section 5 summarizes the work.

### 1.1. State-of-the-Art

In the last two decades, the academic and transportation research community has been working toward the implementation of a driver state monitoring system. For safe driving, it is important to keep track of the driver’s state to allow detection of when short-term driving performance deviates from its normal state. The experimental and commercial implementations often consist of multiple devices that contribute to the goal of valid and reliable evaluation of drivers’ states, such as cognitive distraction, cognitive load, mental fatigue and emotions [31]. However, the literature shows that most road accidents happen due to the driver’s cognitive load. There are many reasons for this inattentiveness; one of the main reasons is high cognitive load. Research shows that measurements of eye movements are often used as factors that statistically correlate with the latent concept of mental workload [32].

Eye movements are very informative as a data source for the analysis of human cognition [33] and an essential part of multi-sourced big driving data for monitoring driver performance [34]. The reason for this is that eye movements indicate the focus of visual attention. Eye movements are typically divided into fixations and saccades–when the eye gaze [35] pauses in a certain position, and when it moves to another position, respectively. Eye movement is used to detect which areas are looked at most frequently and which areas have the longest fixations. Today, camera-based eye-tracking systems are used unobtrusively and remotely in real-time to detect drivers’ eye movements. The most common remote eye tracking systems use multiple cameras to give satisfactory results. However, promising results from using only one camera have recently emerged on the market. Single-camera systems are cheaper, easier to operate and easier to install in a vehicle compared to multi-camera systems [36]. It is shown in [37] that a single webcam can detect eye positions with stable accuracy. Considering performance, the accuracy of a multi-camera system is higher than a single-camera system [38]. The report presented in [32] briefly describes five eye-measuring techniques, i.e., GazeTracker, EOG, JAZZ, Smart Eye and Video-based, along with their advantages and disadvantages. GazeTracker uses a head-mounted infra-red source and CCD camera. This eye tracker sends out near-infrared light and it is reflected in the eyes. Those reflections are picked up by the eye tracker’s cameras. Through filtering and calculations, the eye tracker knows where the eyes are looking. The electrooculogram (EOG) is a very simple way of measuring eye movement activity by placing electrodes around the eye. These electrodes do not measure the eye directly, but they pick up the electric activity of the muscles controlling the eyeball.

Another type of eye-tracking methods involves physically attaching a reference object to the eye, using a contact lens [39]. In most cases, the reference object is a small wire coil embedded in the contact lens, which can measure the movement of the eyes. This eye-tracking system is highly intrusive and causes discomfort for the user.

Many eye-tracking methods presented in the literature are developed based on image processing and computer vision techniques. In these methods, a camera is set to focus on one or both eyes and record the eye movement. There are two main areas investigated in the field of computer vision-based eye tracking: eye detection or eye localization in the image [40] and eye-tracking [21]. Different pattern recognition techniques, such as template matching, are used for eye-tracking. In [41], principal component analysis (PCA) is used to find principal components of the eye image, and an artificial neural network (ANN) is used to classify the pupil position. A particle filter-based eye-tracking method was proposed which estimates a sequence of hidden parameters to detect eye positions [42]. There are also many other approaches found in the literature for eye-tracking, such as Hough transform-based eye tracking [43] and Haar-based cascade classifiers for eye tracking [44].

Cognitive load can be detected based on four assessment methods: primary task performance measures, secondary task performance measures, subjective measures and physiological measures [45]. Different types of features are extracted for each of the assessment methods and either machine learning algorithms or statistical methods are used for cognitive load classification. For feature extraction, a windowing size is considered at the beginning; however, there is no unique window size that can be used for feature extraction. Literature shows that different sampling frequencies are used for different types of data. In the literature, different sampling frequencies were considered for eye movement feature extraction for cognitive load classification. In [46], there were four secondary tasks and each task was performed for 3 min. Then, features were extracted from the entire 3 min of data, i.e., the sampling frequency was 180 Hz. In [47], a sampling frequency of 15 Hz was considered for calculating eye movement parameter fixation duration. The authors considered different window sizes, but the best performance appeared when the time window size was 30 s [48]. However, a different opinion is seen in [49]; the author suggested that it might be unnecessary to limit the window size. After all, the sampling frequency for eye movement feature extraction depends on the characteristics of the data such as the number and the duration of secondary tasks.

## 2. Materials and Methods

### 2.1. Data Collection

Thirty-three male participants aged between 35–50 (42.47 ± 4.39 years) were recruited for the study. Only males were chosen to obtain homogeneous groups from the population. The regional ethics committee at Linköping University, Sweden (Dnr 2014/309-31) approved the study and each participant signed an informed consent form. The experiment was carried out using a car simulator (VTI Driving Simulator III) (https://www.vti.se/en/research/vehicle-technology-and-driving-simulation/driving-simulation/simulator-facilities) (Accessed date: 26 November 2021) which is shown in Figure 1.

The approximate driving time was 40 min, including a practice session of 10 min before the actual driving with cognitive load activity. The driving simulation environment consisted of three recurring scenarios: (1) a four-way crossing with an incoming bus and a car approaching the crossing from the right (CR), (2) a hidden exit on the right side of the road with a warning sign (HE), and (3) a strong side wind in open terrain (SW). In the simulation, the road was a rural road with one lane in each direction, some curves and slopes and a speed limit of 80 km/h. As a within-measure study, each scenario was repeated four times during the driving session where participants were involved in a cognitive load task, i.e., a one-back task, or were driving to pass a scenario (baseline or no additional task). Thus, the cognitive load was annotated as cognitive load class ‘0’ for baseline and cognitive load class ‘1’ for the one-back task. The start and end time of each HE, CR and SW were recorded with a no task and one-back task marker. The one-back task was imposed on drivers by presenting a number aurally every two seconds. The participants had to respond by pressing a button mounted on their right index finger against the steering wheel if the same number was presented twice in a row. The scenarios were designed to investigate the adaption of the driver behavior corresponding to the scenario and cognitive task level (i.e., one-back task).

Two recording systems were used to track and record eye activities. The SmartEye eye-tracking system (http://www.smarteye.se) (Accessed date: 26 November 2021) was the primary device that tracked and captured the eye movements of the drivers, which is considered as ground truth in this paper. The second system was a digital camera that captured the driver’s face and upper body. Each driver had the opportunity to agree or disagree that the video recording should be used at seminars or events when signing the informed consent.

### 2.2. Eye-Pupil Detection

The Materials Figure 2 shows a test participant and his detected eye-pupil position. A summary of the eye pupil position detection and extraction through facial images is presented by a flow chart in Figure 3**.**

Initially, video files are converted into images based on the frame size. In Step 2, the face is detected from the video images through a region of interest (ROI) using the Viola and Jones algorithm [50] and, to speed up the face detection to the next consecutive image frames, face tracking is applied using the Kanade-Lucas-Tomasi (KLT) algorithm [51]. Details and a technical description of face detection are presented in our previous article [52]. Several image processing tasks are conducted to detect eye pupil positions in the image frames. First, the extracted facial ROI is converted into grayscale images and then the grayscale images are transformed into binary images, imposing a threshold value of 0.5. In the next level, the binary image is converted into an inverse image. Inverse image helps to find the edges of the face which are formed due to presence of eyes, nose and mouth. A Sobel edge detection method is used for detecting these edges in the inverse image. Then, the goal is to find the eyes; to do this, it is detected whether there is any circular region or not. Finally two circles for eyes are detected which provide the center of the circle or the center of the eye position. For better understanding, Algorithm 1 is provided with a simplified pseudocode is also provided below:
**Algorithm 1.** Pseudocode for extract eye pupil positionsBEGIN    WHILE until reading the last name     FOR frame number = 1        Read current frame      IF face exist in current frame        Detect face          Select ROI          Detect eyes from the ROI        Perform image processing          Calculate eye pupil position        Save the position         ELSE           Read next frame         ENDIF     ENDFOR    ENDWHILEEND

### 2.3. Feature Extraction

For the feature extraction, the raw eye movement signals are divided into fixation and saccade. The signal is fixation when eye gaze pauses in a certain position, and the signal is saccade when it moves to another position. In brief, a saccade is a quick, simultaneous movement of both eyes between two or more phases of fixation in the same direction. Saccade and fixation are calculated from the time series of eye positions’ raw data (X, Y). First, the velocity is calculated based on two adjacent positions and their respective time and all 13 features are calculated. The list of features is extracted using eye positions which are listed in Table 1.

### 2.4. Classification Methods

In this paper, for cognitive load classification, three approaches have been deployed which are the ML approach, DL approach, and ML + DL approach. 

#### 2.4.1. ML Approach

Figure 4 presents a block diagram for the machine learning approach including data processing, data sets preparation, training, validation and classification steps.

(1) *Input Signals* of the approach are considered as eyeT signals recorded by the SmartEye system (i.e., the eye movement data are recorded in (X, Y) format) and the facial video was recorded by Microsoft LifeCam Studio (https://www.microsoft.com/accessories/en-us/products/webcams/lifecam-studio/q2f-00013) (Accessed date: 26 November 2021) and stored in a separate computer. In (2) *Data processing*, the approach focuses on the eye-pupil position extraction through facial images (presented in II (B)) and feature extraction (presented in II (C)). In (3) *Data Set Preparation*, the extracted features are divided into two classes based on the auditory 1-back secondary task by the participants during simulator driving. Based on the tasks performed by the participants, the classes are defined as ‘0’ represents no cognitive load (n-back task) or baseline (i.e., primary driving task) and ‘1’ represents a one-back Task (i.e., secondary task).

A secondary task was imposed six times for each driver while driving in a scenario. The duration of each secondary task was 60 s. There were 12 scenarios and each driver drove 60 s in each scenario. Eye movement parameters are extracted considering a window size of 30 s. Therefore, there are 24 samples in each test subject, where 12 samples belong to class ‘0’ and the rest of the 12 samples belong to class ‘1’. A summary of the samples in each data set is shown in Table 2.

In the (4) *Model classifiers and classification* results, both training and validation tasks are considered. For the training, five ML algorithms, SVM, LR, LDA, k-NN and DT, are deployed based on the extracted features through training and considered as an instance of supervised learning to classify drivers’ cognitive load tasks [53]. For the *validation*, two cross-validation techniques are conducted which are k-fold cross-validation and holdout cross-validation. In the k-fold cross validation, data are partitioned into k randomly chosen subsets of roughly equal size where k-1 subsets are used for the training and the remaining subset is used for validating the trained model.

This process is repeated k times such that each subset is used exactly once for validation. Holdout cross-validation partitions data randomly into exactly two subsets of a specified ratio for training and validation. This method performs training and testing only once, which minimizes the execution time. Then, for the classification, first true positive (TP), false positive (FP), true negative (TN), and false-negative (FN) are calculated for each ML algorithm. Finally, classification accuracy and *F*_1_-score are obtained.

#### 2.4.2. DL Approach

In the deep learning approach, two deep learning architectures are used which are CNN and LSTM networks.

#### 2.4.3. ML + DL Approach

CNN: Most of the existing CNN models consist of a large number of layers; for example, AlexNet has 25 layers, vgg16 has 41 layers, vgg19 has 47 layers and resnet101 even has 347 layers. The greater number of layers means more complex models and it requires more time to process. In this study, a CNN architecture with 16 layers is designed from scratch to classify the cognitive load. It was emphasized to design a CNN model considering a smaller number of layers so that the processing time of the images can be reduced as much as possible. Among the 16 layers, there is one input layer, three convolutional layers, three batch normalization layers, three relu layers, three max-pooling layers, one fully connected layer, one softmax layer and the final layer or output layer. The input layer reads the time series data and passes it into the series of other layers. The design of the CNN architecture and the dimensions of the hyperparameters are presented below in Figure 5.

LSTM: Another deep learning network called LSTM is used to classify the driver’s cognitive load using time series eye movement data. The LSTM network in this study is a type of recurrent neural network (RNN) and it consists of five layers. The essential layers of an LSTM network are a sequence input layer and an LSTM layer. The time-series data are formed into sequences which are fed into the input layer of the network. Figure 6 demonstrates the architecture of a simple LSTM network for time series classification of cognitive load.

The network starts with a sequence input layer followed by an LSTM layer. To predict class labels, the network ends with a fully connected layer, a softmax layer and a classification output layer. The LSTM layer architecture is illustrated in Figure 7.

This diagram presents the flow of the time-series data (x,y) with features (channels) C of length S through an LSTM layer. In the diagram, $h_{t}$ and $c_{t}$ denote the output or the hidden state and the cell state, respectively.

ML + DL Approach: The combination of two ML + DL approaches is also considered to classify the driver’s cognitive load, which are CNN + SVM and AE + SVM.

CNN + SVM: A DL + ML approach considering CNN + SVM is presented in Figure 8. The CNN model presented in Figure 8 is used to extract features automatically from the raw data and the features are then used for cognitive load classification using a machine learning classifier.

In this case, the SVM classifier has been deployed. The automatic feature extraction is performed in the fully connected layer which is the 14th layer of the networks. Then the extracted features are used to train the SVM model where k-fold cross-validation is performed. Finally, the model classifier is deployed for the classification of cognitive load.

AE + SVM: Another DL + ML approach using AE + SVM is applied for the automatic feature extraction from the raw data and the classification of the cognitive load. The AE, in this case, is a stacked AE, which is presented in Figure 9.

The network of this stacked AE is formed by the two encoders and one softmax layer; however, the second encoder is also called the decoder. The number of hidden units in the first and second encoder is 100 and 50, respectively. In this case, the raw data of size (360 × 3600) are fed into the input layer and then the first encoder is trained. Traditionally, the number of hidden layers should be less than the data size. After training the first AE, the second AE is trained in a similar way. The main difference is that the features that were generated from the first AE will be the training data in the second AE. Additionally, the size of the hidden layers is decreased to 50, so that the encoder in the second AE learns an even smaller representation of the input data. The original vectors in the training data had 3600 dimensions. After passing them through the first encoder, this was reduced to 100 dimensions. After using the second encoder, this was reduced to 50 dimensions. Finally, these 50-dimensional vectors are used to train the SVM model to classify the two classes of cognitive load. K-fold cross-validation approach is considered for the validation.

### 2.5. Evaluation Methods

After the implantation of the proposed approach as a proof-of-concept application, several experiments are conducted where several evaluation methods are used. These experiments are mainly the comparisons between the raw signals, features and classification by both eyeT and camera systems. In addition, significant test between the classes and identification of optimal window size are also considered. The mentioned evaluation metrics are (1) cumulative percentage, (2) box plot and (3) sensitivity/specificity analysis.

*The cumulative percentage* is a way of expressing the frequency distribution of the raw data signals. It calculates the percentage of the cumulative frequency within each interval, much as relative frequency distribution calculates the percentage of frequency. The main advantage of cumulative percentage over cumulative frequency as a measure of the frequency distribution is that it provides an easier way to compare different sets of data. Cumulative frequency and cumulative percentage graphs are the same, except the vertical axis scale. It is possible to have the two vertical axes (one for cumulative frequency and another for cumulative percentage) on the same graph. The cumulative percentage is calculated by dividing the cumulative frequency by the total number of observations (n) and then multiplying it by 100 (the last value will always be equal to 100%). Thus, the cumulative percentage is calculated by Equation (1).
CP = (cumulative frequency ÷ n) × 100(1)

A *box plot* (also known as box and whisker plot) is a type of chart often used in explanatory data analysis to visually show the distribution of numerical data and skewness through displaying the data quartiles (or percentiles) and averages. Here, it shows a five-number summary of a set of data: (1) minimum score, (2) first (lower) quartile, (3) median, (4) third (upper) quartile and (5) maximum score. The *minimum score* contains the lowest scores, excluding outliers. The *Lower quartile* shows the 25% of scores that fall below the lower quartile value (also known as the first quartile). The *median* marks the mid-point of the data and is shown by the line that divides the box into two parts (sometimes known as the second quartile). Half the scores are greater than or equal to this value and half are less. *The upper quartile* shows the 75% of the scores that fall below the upper quartile value (also known as the third quartile). Thus, 25% of the data are above this value. *The maximum score* shows the highest score, excluding outliers (shown at the end of the right whisker). *The upper and lower whiskers* represent scores outside the middle 50% (i.e., the lower 25% of scores and the upper 25% of scores). The *interquartile range (or IQR)* is the box plot showing the middle 50% of scores (i.e., the range between the 25th and 75th percentile).

In *sensitivity/specificity* analysis, based on the measurement of the prediction, the predicted response is compared with the actual response and compute the accuracy of each classifier based model in terms of the evaluation matrices sensitivity or recall, specificity, precision, *F*_1_-score, accuracy and ROC AUC [53]. All these matrices are calculated based on the formula given in Equations (2)–(6).
(2)$$\mathrm{Sensitivity}\;\mathrm{or}\;\mathrm{Recall}\;=\frac{\mathrm{TP}}{\mathrm{TP}+\mathrm{TN}}$$
(3)$$\mathrm{Pecificity}\;=\frac{\mathrm{TN}}{\mathrm{TN}+\mathrm{FP}}\;$$
(4)$$\mathrm{Precision}\;=\frac{\mathrm{TP}}{\mathrm{TP}+\mathrm{FP}}$$
(5)$$\;F_{1}\mathrm{-score}\;=2\times \frac{\mathrm{Precision}.\mathrm{Recall}}{\mathrm{Precision}+\mathrm{Recall}}$$
(6)$$\mathrm{Accuracy}\;=\frac{\mathrm{TP}+\mathrm{TN}}{\mathrm{TP}+\mathrm{TN}+\mathrm{FP}+\mathrm{FN}}$$

Another important measurement is the *receiver operating characteristic* (ROC) curve and *area under the curve* (AUC). The ROC curve shows the performance of a classification model at all classification thresholds. This curve plots two parameters: true positive rate (TPR) and false-positive rate (FPR). Lowering the classification threshold classifies more items as positive, thus increasing both FP and TP. The AUC measures the entire two-dimensional area underneath the entire ROC curve, and it ranges in value from 0 to 1. A model whose predictions are 100% wrong has an AUC of 0.0; one whose predictions are 100% correct has an AUC of 1.0.

Two *statistical significance tests*, Wilcoxon’s signed ranked test and and DeLong’s test, are conducted to compare the performance of the models based on ROC curves. The Wilcoxon signed-rank test is a nonparametric test which is used to compare two sets of scores that come from the same participants and z-score and *p*-values are obtained. The Delong test is performed between two models based on ROC curves and *p*-value and z-score of the two curves are obtained; *p* < 0.05 can be seen as a large difference between the two curves. If the z-score deviates too much from zero then it is concluded that one model has a statistically different AUC from the other model with *p* < 0.05.

## 3. Experimental Works and Results

The aim and objective of these experiments are to observe the performance of the camera system compare to the commercial Eye-Tracking (eyeT) system in terms of raw signal comparisons, extracted features comparisons and drivers’ cognitive load classification. The experimental works in this study are four-fold: (1) *comparison between raw signals* extracted both by the camera system and by the commercial eyeT system, (2) *Selection of Optimal Sampling Frequency*, i.e., identification of the sampling frequency that is best for feature extraction and classification, (3) *comparisons between the extracted features* based on the camera system and the eyeT system and, finally, (4) *cognitive load classification and comparisons* between the camera system and the eyeT system.

### 3.1. Comparison between Raw Signals

This experiment aims to determine if the extracted raw signals of the camera system compare to the raw signal extracted from the eyeT system. Here, a visualization of raw signals and a cumulative percentage of the raw signals have been calculated. For the visualization of the raw signals both by the camera system and by the commercial eyeT system, a test subject is randomly selected and saccade and fixation signals are plotted for 200 samples which are presented in Figure 10.

It is observed that the saccade peaks between eyeT and camera signals are identical, and only the amplitude of the fixation of the camera signal is higher than the amplitude of eyeT. This makes it easy for the feature extraction task by the proposed camera system.

The cumulative percentage experiment aims to see the frequency distribution on the raw data extracted from the proposed camera system compare to the eyeT system. To calculate the cumulative percentage, several steps are followed; they are: *Step* (1): the percentage of absolute differences between eyeT and camera raw signals are calculated for each subject considering x and y signals. *Step* (2): Then, the cumulative percentages are calculated for x and y and their average values are considered as a cumulative percentage for a subject. *Step* (3): Finally, the average cumulative percentage for 30 test subjects is calculated. An example of cumP calculation is presented in Table 3 and the average cumP for 30 test subjects is shown in Figure 11.

As can be observed from Figure 11, the cumulative percentages of 80, 90 and 100 are achieved for the absolute differences while considering as a threshold, i.e., 13, 20 and 40, respectively. That means to achieve 100% accuracy of the raw signal extracted from the camera system compared to the eyeT system, the absolute differences between the two raw signals should be 40 as an average value of 30 subjects.

### 3.2. Selection of Optimal Sampling Frequency

Once the raw signals are extracted and compared, the next task is to extract features for the cognitive load classification. To achieve good features and better classification accuracy, the sampling frequency should be the best selection; thus, this experiment aims to identify the best sampling frequency to extract features and cognitive load classification. In this study data of each of the secondary n-back tasks were imposed on the driver for one minute during simulator driving. Three different time windows, i.e., 60 s, 30 s and 15 s, were considered for feature extraction to observe which window size performs the best for ML algorithms, i.e., SVM, LR, LDR, k-NN and DT, considering *F*_1_-score. Table 4 presents the performance, i.e., *F*_1_-score and *Accuracy*, of all five ML algorithms both for eyeT and camera data. It can be observed that the *F*_1_-score and *Accuracy* are better for 30 s sampling frequency than 60 s and 15 s sampling frequencies. It is also observed that k-fold cross validation (i.e., k = 5) achieves higher accuracy than holdout cross validation. Here, from the comparion it is observed that the highest *F*_1_-score and *accuracy* are achieved when the sampling frequency is 30 Hz. Therefore, all the experiments in subsequent sections only consider the data sets of sampling frequency 30 s and k-fold cross-validation.

### 3.3. Comparisons between the Extracted Features

This experiment focuses on the comparisons between the extracted features by the camera system and the eyeT system. Here, *correlation coefficient* is measured between the feature sets to observe the closeness of the features, and then features are compared between the system considering 0-back and 1-back cognitive loads. The correlation coefficient ‘r’ between the features of the eyeT and camera system is presented in Figure 12. In each case, *p* values are 0 (i.e., <0.05) which which means the correlations are significant. 

The highest value of r is 0.95 and the lowest value is 0.82 which indicates that there is a good positive relation between features of the systems.

Statistical comparison between features extracted from 0-back and 1-back classes are conducted to observe if there are any significant differences in cognitive load with non-cognitive load tasks both for eyeT and camera system. Here, four statistical parameters, maximum (MAX), minimum (MIN), average (AVG) and standard deviation (STD), are calculated for all test subjects. Figure 13 presents the average summary of the statistical measurements for the eyeT system. Figure 14 presents the average summary of the statistical measurements for the camera system.

It can be observed in both cases that there are significant differences between 0-back and 1-back, considering all 13 extracted features.

Box plots are presented to see the significant differences of features between 0-back and 1-back. Here, the summary of the comparisons includes (1) first (lower) quartile, (2) median and (3) third (upper) quartile scores. Figure 15 presents box plots for the features extracted by the eyeT system and Figure 16 presents box plots for the features extracted by the camera system.

According to both figures, 0-back features and 1-back features have significant differences considering 1st quantile, 3rd quantile and median values.

### 3.4. Classification Results

This experiment focuses on the robustness of ML and DL algorithms in terms of cognitive load classification. Here, the average classification accuracy for the experiments is observed for five ML algorithms, SVM, LR, LDA, k-NN and DT, and three DL algorithms, CNN, LSTM and AE, both for eyeT and camera features, considering a 30 Hz sampling frequency. K-fold cross-validation was performed for each classifier, where K is 5. Sensitivity and specificity are also calculated for each algorithm. Table 5 presents the classification accuracy, sensitivity and specificity for SVM, LR, LDA, k-NN, and DT, both for eyeT and camera data.

Different hyperparameters were explored to achieve the highest classification accuracy for all ML models. In the SVM model, three kernel functions, i.e., ‘linear’, ‘gaussian’ (or ‘rbf’) and ‘polynomial’ kernel function, were deployed, where the ‘linear’ kernel function was responsible for producing the highest classification accuracy. In the LR model, a function called ‘logit function’ was used for the classification task. In the LDA model, five types of discriminator functions are used, ‘linear’, ‘pseudolinear’, ‘diaglinear’, ‘quadratic’ and ‘pseudoquadratic’ or ‘diagquadratic’, where the best accuracy is achieved using the ‘linear’ discriminator function. In the k-NN model, different value of k is explored where the best one is k = 10 for ‘Euclidean’ distance function. In the DT model, three criterion functions are explored for choosing a split which are ‘gdi’ (Gini’s diversity index), ‘twoing’ for the twoing rule or ‘deviance’ for maximum deviance reduction (also known as cross-entropy). The best was ‘gdi’, where the maximum number of split is 4.

According to Table 5, the highest overall accuracy achieved is 92% for camera data using SVM classifiers and the highest classification accuracy for eyeT data achieved is 92% for SVM, LR and LDA classifiers which are shaded in gray color.

For the visualization of the tradeoff between true positive rate (TPR) and false-positive rate (FPR), ROC curves are plotted for all ML algorithms and AUC values are calculated which is presented in Figure 17 for eyeT system and Figure 18 for the camera system. In the ROC curve, for every threshold, TPR and FPR are calculated and plotted on one chart.

The higher TPR and the lower FPR are for each threshold is considered as better performance and so classifiers that have curves that are more top-left-side are better. To get one number that tells how good the ROC curve is, the area under the ROC or ROC AUC score is calculated. Here, in the figures, the more top-left the curve is the higher the area and hence higher ROC AUC score. Figure 17 shows that the AUC values for the eyeT system considering SVM, LR and LDA are 0.97 and for k-NN and DT are 0.95 and 0.92, respectively which indicates that the ROC curves for SVM, LR and LDA show better performance than k-NN and DT.

Figure 18 shows that the AUC values for the camera considering SVM, LR and LDA are 0.92 and for k-NN and DT are 0.91 and 0.85, respectively, which indicates that the ROC curves for SVM, LR and LDA show better performance than k-NN and DT.

### 3.5. Statistical Significance Test

Two statistical significance tests (i.e., Wilcoxon test and delong’s test) are conducted between camera and eyeT data for each model. Initially the values of P, H and stats are calculated for each model considering actual and predicted classes of the model using Wilcoxon signed rank test. Here, P is the probability of observing the given result, H is the hypothesis which is performed at the initial hypothesis setting 0.05 (H = 0 indicates that the null hypothesis (“median is zero”) cannot be rejected at the 5% level, H = 1 indicates that the null hypothesis can be rejected at the 5% level) and stats is a structure containing one or two fields (The field ‘signedrank’ contains the value of the signed rank statistic for positive values in X, X-M or X-Y. If P is calculated using a normal approximation, then the field ‘zval’ contains the value of the normal (Z) statistic.).

For conducting this test, initially, the null hypothesis is set to $H_{0}$ that there is no difference in performance measures of a classifier with significance level 0.05. The model/models which is/are significantly different than others are shaded in gray color.

The summaries of the two statistical significance tests for Wilcoxon’s signed ranked and Delong’s test are presented in Table 6 and Table 7, respectively.

## 4. Discussion

The main goal of this study was to investigate the classification accuracy of drivers’ cognitive loads based on saccade and fixation parameters. These parameters are extracted from eye positions throughout the facial images recorded by a single digital camera. The classification performance of the camera system was also compared with the eyeT system to investigate the closeness of the classification results. Based on the literature study, 13 eye movement features are extracted from both the camera and the eyeT data, which are presented in Table 1. These features have shown good performance in cognitive load classification in [9,47] which is also true for this study, where the highest classification for both the eyeT and camera system is 92% which is at least 2% higher than the state-of-the-art accuracy. In [50], the highest cognitive load classification accuracy was achieved at 86% using an ANN algorithm considering a different workload situation. The authors of [51] reviewed the current state of the art and found that the average classification accuracy of cognitive load is close to 90% when considering eye movement parameters. In [52], the highest cognitive load classification accuracy achieved was 87%.

As the raw signal for eye movement was extracted from facial images captured by a camera system, the raw signals were plotted both for eyeT and camera system to observe the closeness of the signals and also to observe the characteristics of the signals, which is shown in Figure 10 which indicates that both the signals from eyeT and camera systems look similar considering saccade peaks and with small differences in the amplitudes of fixation. The actual reason for this amplitude difference of fixation is unknown. However, it might have occurred due to a change in the sampling frequency of eyeT from 50 Hz to 30 Hz. A cumulative percentage between raw signals of eyeT and camera systems is also calculated and presented as another experiment to see the similarity of the two raw signals. Here, the similarity is observed by using a threshold value of absolute difference which is shown in Figure 11 which shows that the cumulative percentage of 80 and 90 is achieved when the absolute differences are 13 and 20 respectively.

Before conducting feature extraction, comparisons and the classification task, an experiment is conducted based on *F*_1_-score and *accuracy* to find the optimal sampling size for feature extraction. As such, three feature sets are generated both from the raw signals using the eyeT and camera systems. Here, the considered sampling sizes are 60 s, 30 s and 15 s. The summary of results using ML algorithms is presented in Table 4. 

Before conducting classification, three types of statistical analyses have been conducted for saccade and fixation features between eyeT and camera systems using several statistical parameters such as correlation coefficients, MAX, MIN, AVG, STD and boxplot. The 1st experiment is conducted on the comparisons between the features of eyeT and camera systems using correlation coefficients. Here, the closeness of the extracted features between the two systems is observed and presented in Figure 13. The results show that the correlation coefficients for all saccade and fixation features between eyeT and camera range from 0.82 to 0.95, which indicates a strong positive relation between eyeT and camera. The 2nd experiment is conducted to observe if there are any significant differences between the features of 0-back and 1-back cognitive load classes. Here, statistical parameters MAX, MIN, AVG and STD show that there are significant differences between 0-back and 1-back features both for eyeT and camera, as presented in Figure 14 and Figure 15, respectively. Boxplots for all 13 features between 0-back and 1-back also confirmed that there are significant differences considering 1st quantile, 3rd quantile and median values both for eyeT and camera. These boxplots are presented in Figure 16 for eyeT and Figure 17 for the camera.

Five machine learning algorithms, SVM, LR, LDA, k-NN and DT, have been investigated to classify cognitive load. The summary of the classification results including sensitivity, specificity, precision, *F*_1_-score and accuracy are presented in V. The highest accuracy both for the eyeT and camera systems is achieved at 92% for SVM. In this paper, different kernel functions such as the linear kernel, Gaussian kernel and polynomial kernel functions are investigated, and results show that linear kernel function performs better than other kernels. However, considering polynomial kernels, the training accuracy increases but then the model tends to overfit due to the huge spreading of the data sets. LR and LDA classifiers also show similar performance in binary classification.

To take the advantage of automatic feature extraction, three DL algorithms CNN, LSTM and AE, are deployed. The results suggest that both DL and DL + ML approaches outperform in a similar manner. However, the highest accuracy of 0.91% is obtained by CNN which is 0.01% higher than LSTM and AE. In CNN, there are few pooling layers for which the features are organized spatially like an image, and thus downscaling the features makes sense, while LSTM and AE do not have this advantage. However, considering the processing time of each image, DL-based technology can be a potential contributor in advanced driver assistive systems. In the experiment, it was noted that the processing time of 1000 images is less than 0.1 sec using an NVIDIA GPU.

For the visualization of the tradeoff between TPR and FPR, ROC curves are plotted and AUC values are calculated. Figure 17 shows that the AUC values for the eyeT system considering SVM, LR and LDA show better performance than k-NN and DT. Figure 18 shows that the AUC values for the camera system considering SVM, LR, and LDA show better performance than k-NN and DT. However, to compare the performance of the models with each other, two statistical significance tests, Wilcoxon signed rank test and Delong’s test, are deployed. For both tests, *p*-values and z-scores resemble similar characteristics. The results based on *p*-values and z-scores suggest that the DT models are significantly different than other models by the null hypothesis *p* < 0.05. Technically, this experiment has several limitations.

## 5. Conclusions

A non-contact-based driver cognitive load classification scheme based on eye movement features is presented, considering a driving simulator environment, which is a new technique for advanced driver assistive systems. The average highest accuracy for camera features is achieved at 92% by using the SVM classifier. In this paper, saccade and fixation features are extracted from the driver’s facial image sequence. However, three DL models are used for automatic feature extraction from raw signals, for which the highest classification accuracy is 91%. It is observed that manual feature extraction provides 1% better accuracy than automatic feature extraction. Non-contact-based driver’s cognitive load classification can be optimized by minimizing extraction error of eye movement parameters, i.e., saccade and fixation features from facial image sequences. Accurate eye detection and tracking is still a challenging task, as there are many issues associated with such systems. These issues include the degree of eye openness, variability in eye size, head pose, facial occlusion, etc. Different applications that use eye tracking are affected by these issues at different levels. Therefore, those factors need to be considered for further improvement. Additionally, the experiment should be conducted considering real road driving for optimum reliability.

## Figures and Tables

**Figure 1 sensors-21-08019-f001:**
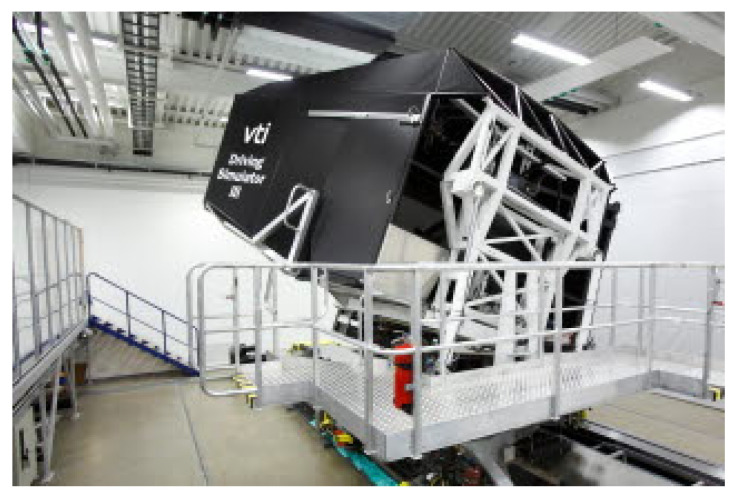
VTI Driving Simulator III (https://www.vti.se/en/research/vehicle-technology-and-driving-simulation/driving-simulation/simulator-facilities) (Accessed date: 26 November 2021).

**Figure 2 sensors-21-08019-f002:**
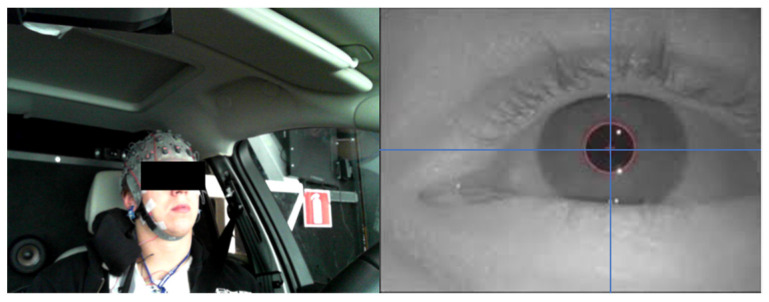
A test participant and his detected eye-pupil position.

**Figure 3 sensors-21-08019-f003:**
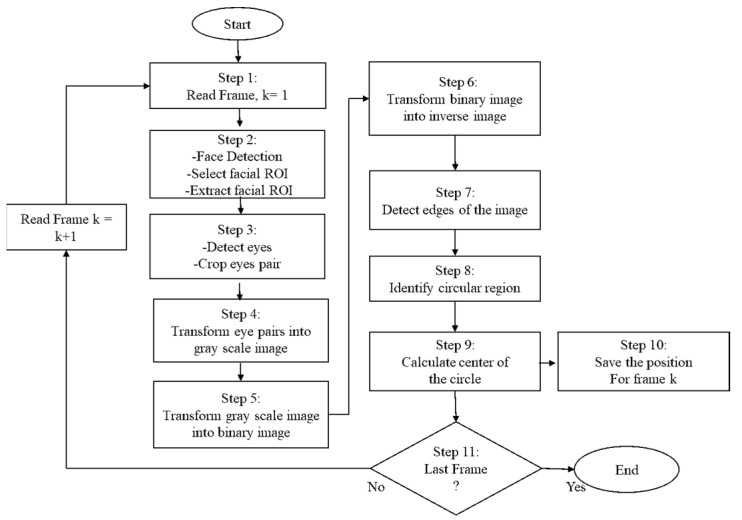
Flow chart of the method for eye positions extraction.

**Figure 4 sensors-21-08019-f004:**
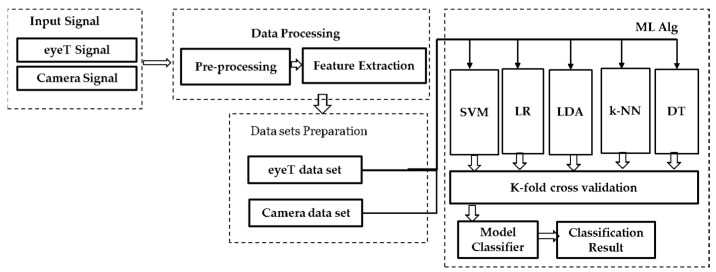
Overview of cognitive load classification machine learning approach: (1) input signal, (2) data processing, (3) data sets preparation, (4) model classifiers, and classification results.

**Figure 5 sensors-21-08019-f005:**
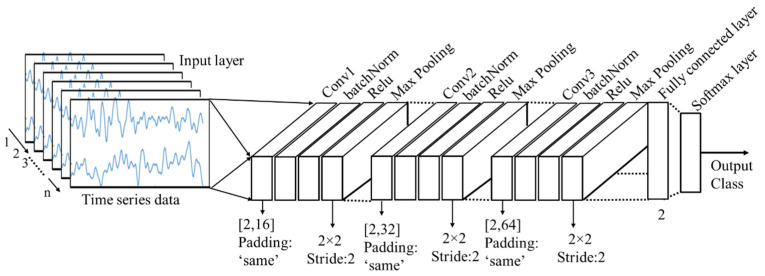
CNN architecture for the time series classification.

**Figure 6 sensors-21-08019-f006:**
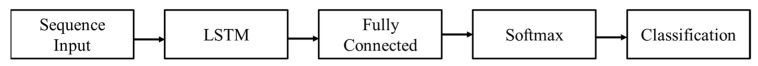
Block diagram of LSTM architecture for the time series classification.

**Figure 7 sensors-21-08019-f007:**
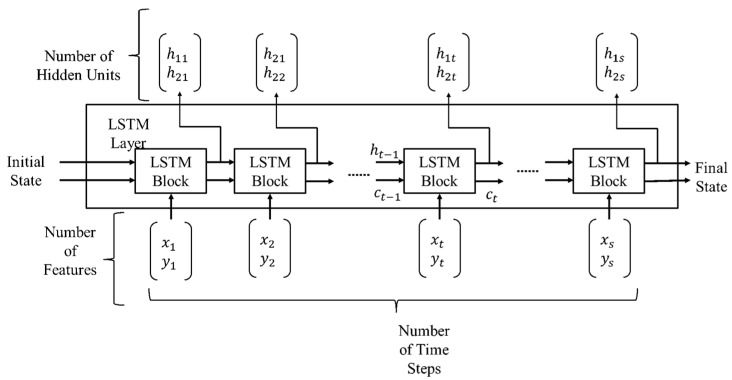
The architecture of LSTM layer.

**Figure 8 sensors-21-08019-f008:**
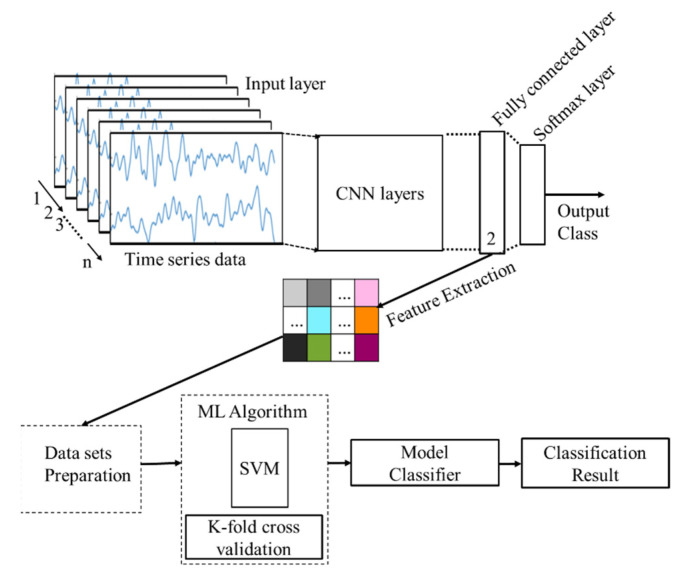
The block diagram of ML + DL approach using CNN + SVM for cognitive load classification.

**Figure 9 sensors-21-08019-f009:**
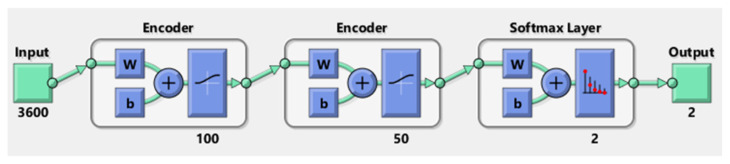
The stacked AE architecture.

**Figure 10 sensors-21-08019-f010:**
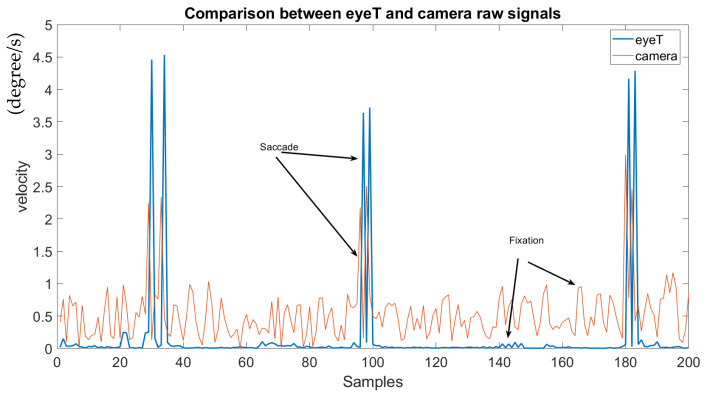
Visualization of the raw signals extracted by the camera system and compared with the eyeT system.

**Figure 11 sensors-21-08019-f011:**
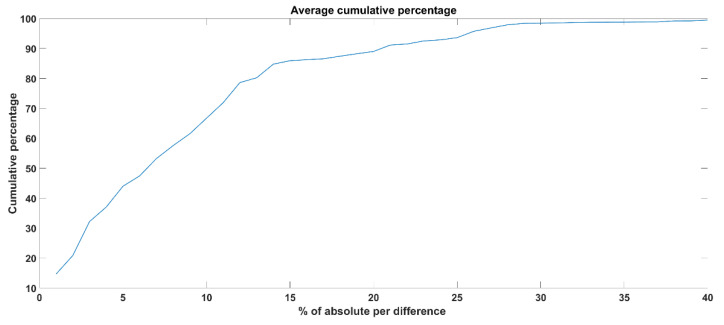
Average cumulative percentage of all 30 test subjects considering raw signals both extracted by the proposed camera system compare to the eyeT system.

**Figure 12 sensors-21-08019-f012:**
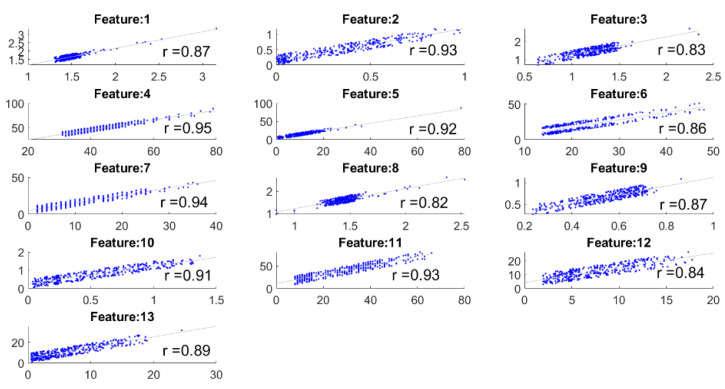
Correlation coefficients between the features extracted both by the eyeT and camera systems.

**Figure 13 sensors-21-08019-f013:**
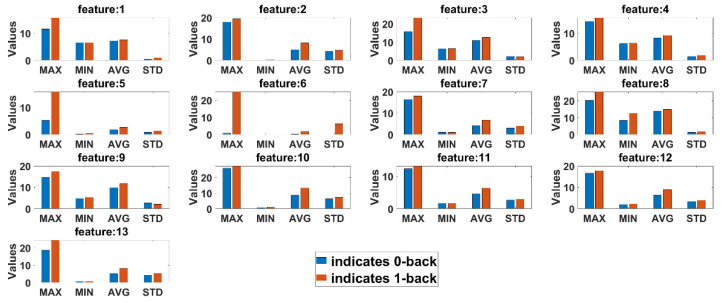
Summary of the statistical parameters comparing n-back task 0 and 1 considering features eyeT system.

**Figure 14 sensors-21-08019-f014:**
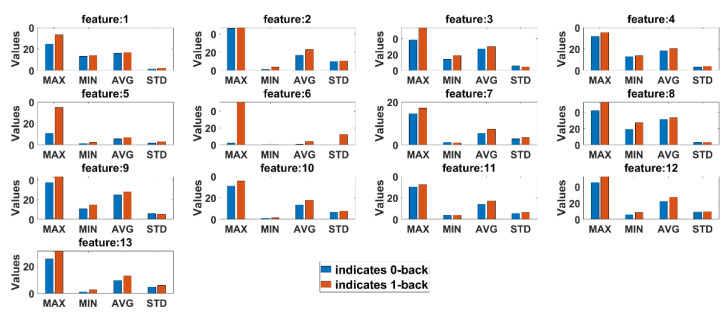
Summary of the statistical parameters comparing n-back task 0 and 1 considering features camera system.

**Figure 15 sensors-21-08019-f015:**
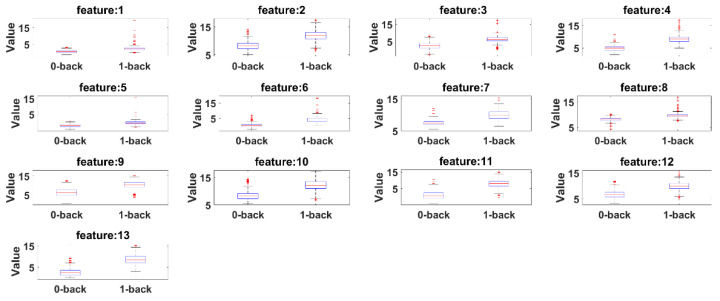
Boxplot of 0-back and 1-back for eyeT features.

**Figure 16 sensors-21-08019-f016:**
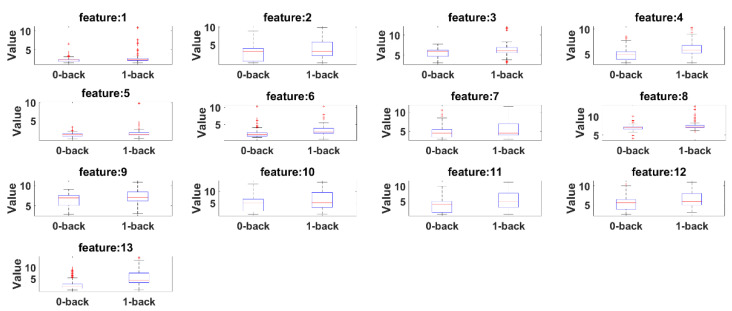
Boxplot of 0-back and 1-back for camera features.

**Figure 17 sensors-21-08019-f017:**
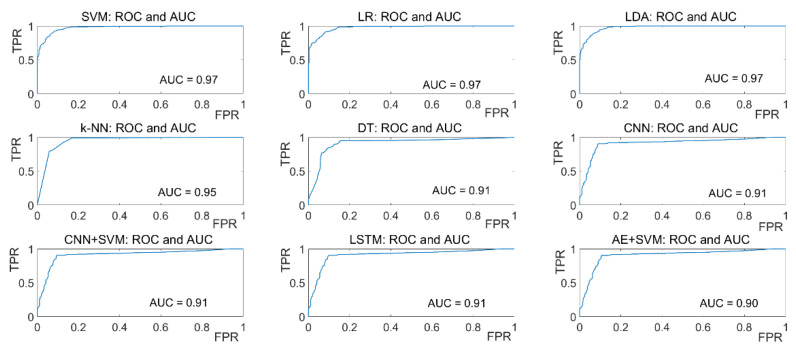
ROC and AUC for ML classifiers considering the eyeT system.

**Figure 18 sensors-21-08019-f018:**
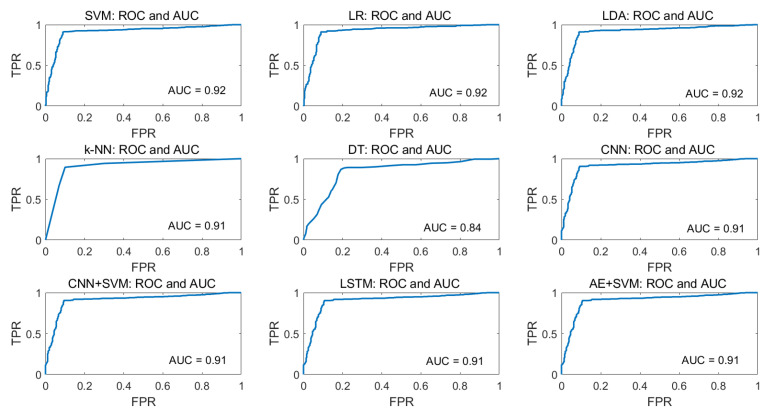
ROC and AUC for ML classifiers considering camera system.

**Table 1 sensors-21-08019-t001:** List of saccade and fixation features.

Types of Features	Feature No.	Name of the Features
Saccade	1	Maximum saccade velocities
	2	Standard deviation of saccade velocities
	3	Average of saccade velocities
	4	Maximum saccade duration
	5	Standard deviation of saccade duration
	6	Average of saccade duration
	7	number of saccades
Fixation	8	Maximum fixation velocities
	9	The standard deviation of fixation velocities
	10	Average of fixation velocities
	11	Maximum of fixation duration
	12	The standard deviation of fixation duration
	13	Average of fixation duration

**Table 2 sensors-21-08019-t002:** Number of cases in eyeT and camera data.

Group	Study	# of Samples in Each Data Set
Set1	eyeT	Class ‘0’: 360 Class ‘1’: 360 Total: 720
Set2	Camera	Class ‘0’: 360 Class ‘1’: 360 Total: 720

**Table 3 sensors-21-08019-t003:** An example to calculate cump.

Sample	eyeT	Camera	abs Diff	% of abs Diff	If (Th <= 15,1,0)	Count	cumP
1	0.143	0.179	0.038	25.635	0	0	0.00
2	0.189	0.179	0.012	6.067	1	1	50.00
3	0.174	0.154	0.015	8.112	1	2	66.67
4	0.183	0.164	0.020	10.530	1	3	75.00
5	0.178	0.180	0.016	8.479	1	4	80.00
6	0.180	0.160	0.007	3.689	1	5	83.33
7	0.170	0.180	0.029	20.214	0	5	71.43
8	0.178	0.180	0.017	8.767	1	6	75.00
9	0.176	0.179	0.015	7.964	1	7	77.78
10	0.176	0.187	0.005	2.640	1	8	80.00

**Table 4 sensors-21-08019-t004:** Summary of the comparison of different sampling frequencies.

ML Model	Cross-Validation Methods	Measurements	Sampling Frequency					
			60 s		30 s		15 s	
			eyeT	Camera	eyeT	Camera	eyeT	Camera
SVM	k-fold	*F*_1_-score	0.8	0.82	0.92	0.92	0.82	0.85
		Accuracy	0.8	0.81	0.92	0.92	0.82	0.85
	Holdout	*F*_1_-score	0.8	0.8	0.91	0.9	0.8	0.83
		Accuracy	0.8	0.8	0.91	0.9	0.8	0.83
LR	k-fold	*F*_1_-score	0.81	0.77	0.92	0.91	0.84	0.85
		Accuracy	0.81	0.78	0.92	0.91	0.84	0.85
	Holdout	*F*_1_-score	0.8	0.79	0.91	0.9	0.82	0.83
		Accuracy	0.8	0.79	0.91	0.9	0.81	0.8
LD	k-fold	*F*_1_-score	0.8	0.78	0.92	0.91	0.82	0.85
		Accuracy	0.8	0.78	0.92	0.91	0.82	0.86
	Holdout	*F*_1_-score	0.8	0.79	0.91	0.9	0.81	0.83
		Accuracy	0.8	0.78	0.9	0.9	0.8	0.82
k-NN	k-fold	*F*_1_-score	0.86	0.82	0.91	0.91	0.87	0.83
		Accuracy	0.86	0.82	0.91	0.91	0.87	0.83
	Holdout	*F*_1_-score	0.86	0.82	0.9	0.9	0.85	0.82
		Accuracy	0.85	0.82	0.9	0.9	0.84	0.81
DT	k-fold	*F*_1_-score	0.89	0.84	0.89	0.9	0.86	0.85
		Accuracy	0.89	0.84	0.88	0.89	0.86	0.85
	Holdout	*F*_1_-score	0.87	0.84	0.88	0.88	0.85	0.84
		Accuracy	0.88	0.84	0.88	0.88	0.84	0.84

**Table 5 sensors-21-08019-t005:** Sensitivity, specificity, precision, *F*_1_-score, and accuracy for svm, lr, lda, k-nn, and dt classifiers for both eyet and camera data, where total observation is 720, 0-back classes are 360 and 1-back classes are 360.

Data	Algorithms	Criteria								
		True Positive (TP)	False Nagative (FN)	False Positive (FP)	True Negative (TN)	Sensitivity or Recall	Specificity	Precision	*F*_1_-Score	Accuracy
eyeT	SVM	338	36	22	324	0.9	0.94	0.94	0.92	0.92
	LR	331	30	29	330	0.92	0.92	0.92	0.92	0.92
	LDA	335	35	25	325	0.91	0.93	0.93	0.92	0.92
	k-NN	336	44	24	316	0.88	0.93	0.93	0.91	0.91
	DT	322	45	38	315	0.88	0.89	0.89	0.89	0.88
	CNN	325	32	35	328	0.91	0.9	0.9	0.91	0.91
	CNN + SVM	323	31	37	329	0.91	0.9	0.9	0.9	0.91
	LSTM	322	31	38	329	0.91	0.9	0.89	0.9	0.9
	AE + SVM	320	33	40	327	0.91	0.89	0.89	0.9	0.9
Camera	SVM	336	35	24	325	0.91	0.93	0.93	0.92	0.92
	LR	330	34	30	326	0.91	0.92	0.92	0.91	0.91
	LDA	332	35	28	325	0.9	0.92	0.92	0.91	0.91
	k-NN	337	45	23	315	0.88	0.93	0.94	0.91	0.91
	DT	324	40	36	320	0.89	0.9	0.9	0.9	0.89
	CNN	324	32	36	328	0.91	0.9	0.9	0.91	0.91
	CNN + SVM	323	32	37	328	0.91	0.9	0.9	0.9	0.9
	LSTM	317	31	43	329	0.91	0.88	0.88	0.89	0.9
	AE + SVM	318	31	42	329	0.91	0.89	0.88	0.9	0.9

**Table 6 sensors-21-08019-t006:** Summary of the wilcoxon test.

Model	eyeT		Camera	
	z-Score	*p*	z-Score	*p*
SVM	2.11	$9.80\;\times \;10^{-5}$	−0.24	$4.04\;\times \;10^{-4}$
LR	0.53	$7.02\;\times \;10^{-4}$	0.13	$5.51\;\times \;10^{-5}$
LDA	0.38	$6.46\;\times \;10^{-8}$	0.003	$5.02\;\times \;10^{-7}$
K-NN	2.56	$9.95\;\times \;10^{-7}$	−0.23	$4.10\;\times \;10^{-6}$
DT	1.79	$9.63\;\times \;10^{-1}$	1.42	$9.23\;\times \;10^{-1}$
CNN	−0.36	$3.58\;\times \;10^{-5}$	−0.48	$3.15\;\times \;10^{-6}$
CNN + SVM	−0.72	$2.35\;\times \;10^{-6}$	−0.60	$2.75\;\times \;10^{-5}$
LSTM	−0.84	$2.01\;\times \;10^{-4}$	−1.39	$8.20\;\times \;10^{-5}$
AE + SVM	−0.82	$2.07\;\times \;10^{-6}$	−1.28	$9.95\;\times \;10^{-5}$

**Table 7 sensors-21-08019-t007:** Summary of the delong’s test to compare auc values at significant levels 0.05 and comparison of z-scores and *p*-values between models.

Model	eyeT			Camera		
	AUC	z-Score	*p*	AUC	z-Score	*p*
SVM	0.97	−0.36	$6.7\;\times \;10^{-9}$	0.92	−2.98	$5.8\;\times \;10^{-7}$
LR	0.97	−0.29	$1.4\;\times \;10^{-6}$	0.92	−3.41	$2.5\;\times \;10^{-8}$
LDA	0.97	−0.32	$5.2\;\times \;10^{-10}$	0.92	−3.26	$4.7\;\times \;10^{-9}$
K-NN	0.95	−0.25	$8.9\;\times \;10^{-5}$	0.91	−3.01	$6.5\;\times \;10^{-7}$
DT	0.91	1.03	$5.4\times \;10^{-1}$	0.84	1.06	$6.83\times \;10^{-1}$
CNN	0.91	−1.30	$2.1\;\times \;10^{-12}$	0.91	−1.45	$2.5\;\times \;10^{-9}$
CNN + SVM	0.91	−1.52	$3.5\;\times \;10^{-8}$	0.91	−2.03	$4.02\;\times \;10^{-8}$
LSTM	0.91	−1.24	$4.1\;\times \;10^{-11}$	0.91	−1.72	$3.78\;\times \;10^{-11}$
AE + SVM	0.90	−9.82	$2.0\;\times \;10^{-7}$	0.91	−8.79	$1.98\;\times \;10^{-8}$

## Data Availability

The data is not publicly available due to GDPR issue.

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
