# Peer review of "Vision-Based Driver’s Cognitive Load Classification Considering Eye Movement Using Machine Learning and Deep Learning"

_sensors, 2021, doi:10.3390/s21238019_

Round 1
Reviewer 1 Report
The authors propose a vision-based method to extract useful parameters from driver’s eye movement signals using deep learing and machine learning methods. Different feature extraction schemes have also been explored. overall, the topic is interesting, and the paper is well organized. However, there still exists a few issues that have to be addressed before publication.
- The authors claim to achieve 92% classification accuracy for their proposed vision-based driver’s cognitive load classification model. However, no comparative analysis is provided to contrast the achieved outcomes with other studies, and hence and no assumption about the robustness of the envisaged model could be made. It is strongly recommended to compare the proposed method with other state-of-the-art (SOTA) studies and list them in the paper.
- The deep learning-based images classification is time-consuming owing to its deep architecture and a large number of parameters. The authors claim that their method could be a potential contributor in advanced driving assistive systems. What is the single image processing time of the elicited model? How many images per second can the model classify, and what is the per second error rate of the model? Such factors are important, considering the road safety necessitates prompt identification of the anomaly.
- What is the physical significance of the 16 layered CNN utilized in this study? Did the author adopt this architecture from any other SOTA studies or formulate it from scratch? It would be promising to shed light on different layers and their contribution towards classification.
- The machine learning (ML) algorithms have different associated parameters, i.e., k-NN has the number of nearest neighbors and type of distance metric, SVM and LDA depend on the kind of kernel functions, etc. It is advised to use different parameters settings for ML classifiers and showcase the best performing settings.
- Figures 11, 13, 14, 15, 16, and 17 are not clearly visible. Try to increase the window and font size of these figures and use high-resolution images.
- Is it possible to give the proposed framework an explicit name with an abbreviation? It will help better recognize the model rather than memorizing it by the signal processing modules in the framework.
Reviewer 2 Report
This paper presents a computer vision-based method to extract features from driver’s eye movement signals using deep learning architectures which is combined with manual feature extraction based on domain knowledge. The paper must be revised to improve the technical presentation of the methodology and the results.
Comments:
- The novelty and contribution of this study must be explicitly stated.
- Describe the proposed method as an algorithm in pseudocode.
- Describe in more detail how you calculated the features presented in Table 1.
- Did you account for subject fatigue? See “Modelling eye fatigue in gaze spelling task” and discuss.
- Your stacked AE architecture presented in Figure 9 does not have a decoder, so strictly speaking this is not an autoencoder.
- Figure 10: add the units of measurement to the velocity axis.
- Figure 12: I do not see the difference between values. Maybe you can change the range of accuracy values for better visibility and comparability?
- Figure 13: in addition to r values, also present the p-values.
- A small size of subplots in Figures 14-17 does not make them useful in comparing the values.
- The description of the statistical testing procedure does not make clear which pairs of models are compared. I suggest also to use the non-parametric Friedman test and visualize the mean ranks of models using the Critical Difference diagram from the post-hoc Nemenyi test.
- In the discussion section, discuss the limitations of your approach.
Round 2
Reviewer 1 Report
The authors have successfully addressed all my concerns, and thus, I would like to recommend its publication in Sensors.
Reviewer 2 Report
All my comments were addressed and the manuscript was well revised.